# Biased Pressure: Cyclic Reinforcement Learning Model for Intelligent Traffic Signal Control

**DOI:** 10.3390/s22072818

**Published:** 2022-04-06

**Authors:** Bunyodbek Ibrokhimov, Young-Joo Kim, Sanggil Kang

**Affiliations:** 1Department of Computer Engineering, Inha University, Inha-ro 100, Nam-gu, Incheon 22212, Korea; bunyod.ibrokhimov@gmail.com; 2Electronics and Telecommunications Research Institute (ETRI), 218 Gajeong-ro, Yuseong-gu, Daejeon 34129, Korea; kr.yjkim@etri.re.kr

**Keywords:** reinforcement learning, intelligent traffic signal control, optimization

## Abstract

Existing inefficient traffic signal plans are causing traffic congestions in many urban areas. In recent years, many deep reinforcement learning (RL) methods have been proposed to control traffic signals in real-time by interacting with the environment. However, most of existing state-of-the-art RL methods use complex state definition and reward functions and/or neglect the real-world constraints such as cyclic phase order and minimum/maximum duration for each traffic phase. These issues make existing methods infeasible to implement for real-world applications. In this paper, we propose an RL-based multi-intersection traffic light control model with a simple yet effective combination of state, reward, and action definitions. The proposed model uses a novel pressure method called Biased Pressure (BP). We use a state-of-the-art advantage actor-critic learning mechanism in our model. Due to the decentralized nature of our state, reward, and action definitions, we achieve a scalable model. The performance of the proposed method is compared with related methods using both synthetic and real-world datasets. Experimental results show that our method outperforms the existing cyclic phase control methods with a significant margin in terms of throughput and average travel time. Moreover, we conduct ablation studies to justify the superiority of the BP method over the existing pressure methods.

## 1. Introduction

Traffic congestion has become increasingly concerning in recent years, having a huge impact on the economy of major cities and countries. According to Forbes, traffic congestion costs $74.1 billion annually in the freight sector alone in the United States [1]. Urban cities suffer 149 h lost a year, on average, making it 24.4 min per day. The cost of traffic congestion in South Korea surpassed 50 trillion KRW (as of 2017) [2], which makes traffic congestion one of the primary urban issues. Reducing congestion would have significant benefits, not only for the country’s economy but also for its environmental welfare by mitigating millions of kilograms of harmful CO_2_ emissions and for its societal interest by increasing productivity at workplaces [3,4]. Among the known factors in traffic, such as increasing demand for transportation and the design of road structures, traffic light control plays a major role in improving traffic management.

Traditional traffic signal control systems rely on manually-designed traffic signal plans, such as SCATS [5] and SCOOT [6] systems. The traffic signals use pre-defined fixed time duration in each cycle for the given intersection, calculated based on historical information of the intersection. Starting time and duration of green and red lights change depending on different times of the day, e.g., flat traffic hours or rush hours. Later works proposed more adaptive and intelligent ways of adjusting phase cycle and durations by measuring traffic demand fluctuations and volume-to-capacity ratio [7] or by optimizing offsets to decrease the number of stops for vehicles [8,9,10]. The latter method is popularly known as GreenWave. Although such transportation engineering methods perform well for the routine traffic volumes, they falter during unexpected events (also known as non-recurring congestion events), such as disruptions or/and accidents in neighboring intersections, weather condition changes, constructions in certain parts of the city, and popular public events such as football events. In order to deal with such non-recurring congestion events, traffic signal control systems should employ intelligent algorithms which are responsive to dynamic and real-time traffic flow. With the development of artificial intelligence (particularly, adaptive optimization models and deep neural networks) and navigation applications (e.g., Google maps) over the years, vehicle trajectory data and traffic data can be collected and used to optimize traffic control systems. Among many, some nature-inspired optimization algorithms, such as Ant Colony Optimization (ACO) [11,12] and Artificial Bee Colony (ABC) [13] approaches, were employed to minimize the travel time of vehicles. The ABC algorithm usually uses meta-heuristic dynamic programming tools to determine the green light duration and is inspired by the behavior of honey bees. Similarly, the ACO algorithm is based on the behavior of ants in finding optimal paths to their food. The ACO employs the principle of depositing pheromones to determine optimal paths for vehicles to improve traffic movements. However, these methods are mostly used for finding the shortest/optimal path for vehicles rather than essentially adjusting traffic signals according to the intersection load. In recent years, many researchers have studied reinforcement learning (RL) techniques to dynamically analyze real-time traffic and adjust traffic light signal parameters (phase selection, cycle length, and duration of phase) according to observed information [14,15,16]. RL is widely used in the context of smart traffic light control theory and its applications because RL can facilitate traffic systems to learn and react to various traffic conditions adaptively without the need for fixed traffic programs and human labor [17]. A simplistic representation of the RL model is shown in Figure 1. An *agent* of the RL model operates as an experienced policeman who manages the intersection by observing the *environment* (e.g., the number of cars in each segment of intersection) and takes *action* (e.g., chooses the best signal phase) by waving special signals to drivers. The objective of an agent is to increase cumulative *reward* (e.g., the number of cars that passed the intersection).

In RL problems, defining the state representation, action space, and reward is an important task, and according to this definition choice, many studies differ from each other. In earlier stages, RL models were proposed for single intersections using tabular Q-learning. Later, deep RL-based models that can represent more complex states are proposed [18,19,20,21]. However, there are some limitations to evaluate the performance of single intersections models. For example, even though the reward gained by the model is high, the outcome of this intersection changes the flow of other intersections. As a result, vehicles passed from one intersection may cause traffic congestions in the next intersection. To address this issue, several researchers proposed RL-based multi-intersection (e.g., multi-agent) models. Although they showed promising results in simulation conditions, most of the existing methods cannot be implemented in real-world applications due to (1) negligence towards real-world constraints, such as minimum/maximum duration for signal phases and maintaining the phase order, and (2) complex representation of traffic situation, i.e., high-dimensional state definition or complicated action definition. As for (1), most of the recent research works [16,22,23,24] are based on non-cyclic control, i.e., phase order (also called phase sequence) is not guaranteed in order to achieve absolute flexibility. Even though this type of approach contributes to maximizing the throughput of the intersection, such model designs lead to *starvation* in other lanes and restrict the movement of pedestrians. Moreover, irregular phase switches lead to confusion and frustration of drivers, which may result in accidents and/or dangerous situations. As for (2), research works [16,20,25] use complex matrix-based state representation and images of vehicle positions in the incoming lanes of the intersection, which cannot be implemented in large-scale real-world applications due to computational costs and application latency. Therefore, new methods with lightweight state, action, and reward definitions are required for cyclic phase traffic light control systems.

In this paper, we propose a cyclic phase RL model for traffic light control. We introduce a new coordination method called Biased Pressure (BP) that includes both the phase pressure and the number of approaching/waiting vehicles associated with the phase. We use an advantage actor-critic (A2C) method to take advantage of both value-based [26] and policy-based [27] RL methods (for more details, refer to Section 3). Moreover, our proposed model considers the above-mentioned real-world constraints in the model design and implementation. We test our model in both synthetic and real-world datasets and compare its performance with the related methods. Our contributions in this research are summarized as follows:We propose a scalable multi-agent traffic light control system. Decentralized RL agents are used to control traffic signals. Each agent makes decisions based on its own observation. Since neighboring intersections or other intersections in the road network do not negotiate to make a decision, we can achieve a scalable model.We introduce a BP method to determine the phase duration of the traffic signal. BP is an optimized version of the *pressure* method from transportation engineering, which aims to maximize the throughput of an intersection. BP is especially useful when the action definition is based on cyclic phase control.We maintain *must-have* constraints of the traffic signal plan in the definitions of state, action, and reward function to make our method feasible for real-world applications. Our state and reward definitions are simple yet effective in design and do not depend on the number of traffic movements so that our method can be applied to different road structures with multiple allowed traffic movements.

We believe this is the first work including a combination of the above-mentioned contributions.

The remaining structure of this paper is as follows. Section 2 discusses conventional and RL-based related work in the context of traffic light control. The background of the RL algorithm is introduced in Section 3. Section 4 explains our methodology and its learning mechanisms, agent design, and network design in detail. Section 5 describes the experimental environment, evaluation metrics, datasets, and compared methods. Section 6 demonstrates ablation studies on the BP method, extensive experiment results, and a comparison of our method with existing studies and methods. Finally, Section 7 concludes the paper.

## 2. Related Work

### 2.1. Conventional Traffic Light Control

Conventional traffic light control systems can be divided into two main groups: fixed-time traffic signals methods and optimization-based transportation methods. The fixed-time traffic signal usually uses historical data to design traffic signal plans for each intersection. This traffic signal plan often includes a fixed duration for each phase, cycle length, and a fixed time interval (offset) [5,6,28]. Using prior information of the intersection (e.g., traffic movement data, daily average flow), these parameters can be determined, and often separately for peak and rush hours. The optimization-based transportation methods initially start with pre-timed cycle lengths, offsets, and phase split and gradually optimize the cycle length and phase split based on traffic demand fluctuation information. For example, Koonce et al. [7] employed the Webster method to adjust phase durations by measuring fluctuation information and volume-to-capacity ratio for a single intersection. Moreover, this method also uses critical lane volumes to optimize traffic flow served by each lane. However, the method is applied to optimize parameters for a single intersection. To improve coordination between several (multiple) intersections, the GreenWave [8,9,10] method was proposed. This method determines the offsets between the beginning of green lights in the consecutive intersections so that vehicles passed from one intersection would not cause traffic congestion in the next intersection. By reducing the number of stops of vehicles, the method achieves a shorter travel time and, thus, produces higher throughput. Moreover, this traffic plan changes during rush hours and/or special days of the week or month, which gives more flexibility and optimization. The GreenWave method, however, gets a lot more challenging due to dynamic traffic flows and volume coming from opposite directions, as it only optimizes flow for unidirectional traffic, and therefore, it is not flexible for non-recurring traffic situations.

Some other optimization-based methods are proposed to minimize vehicle travel time. For example, Maxband [29] was developed to minimize the number of stops of vehicles along two opposite arterials to improve the efficiency of the traffic flow. By finding a maximal bandwidth based on the signal plan, more traffic can progress through the intersections without stops. However, Maxband requires all intersections to have the same cycle length. In comparison, the self-organizing traffic light control (SOTL) [30,31] method determines whether to keep or change the current phase based on the requests from current and other competing phases. For example, SOTL changes to the next phase if there is a request from the competing phase and the current phase’s green light is larger than the hand-tuned threshold. Otherwise (i.e., when there is no request from other phases or if the current phase is still ongoing), it keeps the current phase. Moreover, the request from the other phase is generated when the number of waiting vehicles at the red light is larger than the threshold. Later, the MaxPressure [32,33] concept was introduced to maximize the throughput of the road network by pressing the traffic flow at the intersection. The aim of this method is to balance the queue length of adjacent intersections by minimizing the pressure of the phase. If the pressure of each intersection is minimized, then the maximum throughput is achieved for the whole road network. The MaxPressure method has become a popular approach in the transportation field as it can be integrated with modern deep learning methods, and is often used as a baseline for state-of-the-art methods.

### 2.2. RL-Based Traffic Light Control

In contrast with conventional methods, RL-based traffic light control methods directly learn and optimize traffic signal plan by interacting with the environment. Such RL methods are divided into two major groups: (1) single intersection traffic light control where the RL agent monitors and optimizes a single intersection [16,18,19,20,21] and (2) multi-intersection traffic light control where the RL agent monitors multiple intersections in the area [24,25,34,35,36,37,38,39]. Since the real-world scenarios at the city-level include numerous intersections, multi-intersection traffic light control systems are preferred over single intersection control systems.

In addition, RL methods can be categorized further according to the state definition of the agent. For example, in research works [21,36,40], the positions of vehicles were used to represent the state of the intersection. In this approach, an information matrix can be used to update the position of each vehicle. Some methods [16,17,23,40,41,42,43] use queue length as their state definition, which is simpler than tracking the position of each vehicle. Recent state-of-the-art methods [25,35,39] use *pressure* of the intersection to define the state. The pressure approach considers the number of vehicles in both incoming (i.e., entering vehicles) and outgoing (i.e., exiting vehicles) lanes of the intersection, and it is calculated by subtracting the number of exiting vehicles from the number of entering vehicles. Although one state representation is not necessarily ‘better’ than the other in terms of performance, a simpler approach to define the state is preferred because complex states, such as vehicle position, create undesired computational cost and lead to latencies, especially in the large-scale road networks.

RL-based traffic light control systems can also be differentiated by their action definition. Two types of action definition, namely phase selection (i.e., which phase to set) and phase duration selection (i.e., what duration to set for the next phase), are commonly used in recent studies. Most research works, including [16,23,24,25,34,39], use phase selection, in which an agent tries to select the optimal phase according to the traffic state. In this action definition, the order of the phase and the overall cycle length is not considered. At each time interval, an agent selects an optimal phase to increase its cumulative reward. Even though this approach produces high efficiency, it cannot be implemented in most real-world intersections because the phase order and minimum/maximum phase duration are not guaranteed. Moreover, phase selection-based methods tend to change traffic light signal many times in a short period of time which can confuse drivers in the real world. This tendency has also been identified by [35,44]. To solve this problem, another group of methods [17,21,35] uses phase duration selection, where an agent selects the optimal duration for the next phase upon switching. In [21], the phase duration is adjusted by increasing or decreasing it by 5 s. However, this method changes the phase duration relatively slower; thus, it cannot respond to dramatic traffic changes. In [35], the cycle length is fixed, and the duration of each phase is selected proportionally which adds up to the fixed cycle length. In [17], the action space of possible durations for phases is directly defined, and an agent selects the optimal duration from this action space. However, none of these methods offer an optimal combination of state, reward, and action definitions. In this paper, we present lightweight state and reward definitions using an improved pressure method called Biased Pressure, and our action definition guarantees phase order as well as the minimum/maximum duration for each phase to maintain real-life constraints.

## 3. Background of Deep RL Algorithm

Deep RL is one of the machine learning algorithms that combines RL and deep learning. In RL, an agent interacts with the environment and learns a “good behavior” to maximize the objective reward function using trial and error *experience*. Based on this experience, the deep RL agent further analyzes its environment through *exploration* and learns to take better action at each time step. In recent years, deep RL has become progressively popular in many machine learning domains and real-world applications, such as robotics [45], finance [46], self-driving cars [47], healthcare [48], smart grids [49], and social networks [50], even beating human performances in some domains. This progressive leap can be seen, especially, in game theory, where deep RL agents successfully beat the world’s top players in Poker, Go, and Chess [51,52,53]. This type of learning process is regarded as Markov Decision Process (MDP) and can be defined using five-tuple $<S,\;A,\;R,\;T,\;\gamma >$, where:

S is the state space,A is the action space,R is the reward function,$T:S\times A\times S\to \left[0,1\right]$ is the transition function,γ∈ [0,1) is the discount factor.

At each time step t, an agent receives the current state $s_{t}$ and reward $r_{t}$. Then, it takes an action $a_{t}$, which results in state transition $s_{t+1}$ and reward $r_{t+1}$, determined by $\left(s_{t},a_{t},s_{t+1}\right)$. The objective of the agent is to learn a policy $\pi \left(s,a\right):A\times S\to \left[0,1\right]$, which maximizes the expected cumulative reward. Generally speaking, π is a mapping states $s_{t}$ to action $a_{t}$. This process continues until an agent reaches a terminal state, e.g., the end of the game.

The expected reward can be represented with the *Q*-value function $Q^{\pi }\left(s,a\right)$, which is defined as Equation (1):(1)$$Q^{\pi }\left(s_{t},a_{t}\right)=E\left[\overset{\infty }{\underset{k=0}{\sum }}\gamma^{k}r_{t+k}\;|\;s_{t}=s,\;a_{t}=a,\;\pi \right]$$

The intuition behind the equation is that the agent tries to get maximum reward by taking action $a_{t}$ in the current state $s_{t}$, following policy π. The action policy can be obtained recursively, as shown in Equation (2):(2)$$Q^{\pi }\left(s_{t},a_{t}\right)=\underset{s^{t+1}\in S}{\sum }T\left(s_{t},a_{t},s_{t+1}\right)\left(R\left(s_{t},a_{t},s_{t+1}\right)+\gamma Q^{\pi }\left(s_{t+1},a_{t+1}\right)\right)$$
and the optimal *Q*-value function $Q^{*}\left(s_{t},a_{t}\right)=\underset{\pi \in \Pi }{\max}Q^{\pi }\left(s_{t},a_{t}\right)$ can be defined as Equation (3):(3)$$Q^{*}\left(s_{t},a_{t}\right)=\underset{s^{t+1}\in S}{\sum }T\left(s_{t},a_{t},s_{t+1}\right)\left(R\left(s_{t},a_{t},s_{t+1}\right)+\gamma \;\underset{a_{t+1}}{\max}Q^{*}\left(s_{t+1},a_{t+1}\right)\right)$$

Then, the optimal policy can be derived directly from $Q^{*}\left(s_{t},a_{t}\right)$, as shown in Equation (4):(4)$$\pi^{*}\left(s_{t}\right)=\underset{a\in A}{\mathrm{argmax}}Q^{*}\left(s_{t},a_{t}\right)\;$$

The intuition behind obtaining optimal policy is selecting the best action which maximizes the expected reward. In many cases, action space available to the RL agent is limited or/and restricted. Therefore, the agent needs to learn how to map the state space into an action space. In this scenario, the agent might have to think about the long-term outcome of its actions, i.e., maximizing future gain, even though the current action may result in a smaller or even negative reward. The importance of immediate and future reward can be defined by the discount factor γ.

In value-based methods, e.g., *Q*-learning [26], an agent relies on value function optimization, shown in Equation (1), without an explicit policy function. Its parameter update is based on one-step temporal difference sampled using agent experience stored in experience replay in the form of $<s_{t},a_{t},r_{t},\;s_{t+1}>$:(5)$$Y_{j}^{Q}=r_{t}+\gamma \;\underset{a_{t+1}}{\max}Q^{*}\left(s_{t+1},a_{t+1};\;\theta_{j}\right)$$
where $\theta_{j}$ denotes parameters at the jth iteration. After each iteration, the parameters of θ are updated by minimizing the loss function (temporal difference) $L\left(\theta \right)=Y_{j}^{Q}-Q\left(s,a;\theta_{j}\right)$:(6)$$\theta_{j+1}=\theta_{j}+\alpha \left(Y_{j}^{Q}-Q\left(s,a;\theta_{j}\right)\right)$$
where α denotes a learning rate. This learning mechanism approximates $Q\left(s,a;\theta_{j}\right)$ towards $Q^{*}\left(s_{t},a_{t}\right)$ from Equation (3) after many iterations, given the assumption that the experience gathered (through exploration) in experience replay is sufficient.

Policy-based methods, e.g., REINFORCE [54], directly optimizes policy function from the parameterized model $\pi_{\theta }$. In such methods, an agent selects the optimal policy $\pi_{\theta }$ that maximizes the expected return. Policy evaluation $Q^{\pi }\left(s_{t},a_{t}\right)$ is used to evaluate policy improvement, which increases the likelihood of the actions to be chosen by the agent. Policy gradient estimator $\nabla_{\omega }L\left(\omega \right)$ is derived by Equation (7):(7)$$\nabla_{\omega }L\left(\omega \right)=E\left[\nabla_{\omega }\log\pi_{\omega }\left(s,a\right)\;Q^{\pi_{\omega }}\left(s,a\right)\right]$$
where ω denote policy parameters and the term $\nabla_{\omega }\log\pi_{\omega }\left(s,a\right)$ is derived from the likelihood ratio trick $\nabla_{\omega }\log\pi_{\omega }\left(s,a\right)=\frac{\nabla_{\omega }\pi_{\omega }\left(s,a\right)}{\pi_{\omega }\left(s,a\right)}$, as in $\nabla \log x=\frac{\nabla x}{x}$. Finally, parameters are updated with a learning rate of $\alpha_{\pi }$, as shown in Equation (8):(8)$$\omega_{j+1}=\omega_{j}+\alpha_{\pi }\;\nabla_{\omega }\log\pi_{\omega }\left(s,a\right)\;Q^{\pi_{\omega }}\left(s,a\right)$$

Although such policy-based methods are more robust to nonstationary MDP transitions (because they use cumulative reward directly, instead of estimating at each iteration, Equation (7)), they experience a high variance in log probabilities and cumulative return due to random sampling. This results in noisy gradients, which cause unstable learning. Variance, however, can be reduced by using the advantage actor-critic (A2C) method [55]. A2C method consists of an actor which takes an action by following a policy and a critic which estimates the value function of the policy. Generally speaking, A2C takes the advantage of both policy-based and value-based methods. The term ‘advantage’ in the definition of A2C refers to an advantage value and it is calculated as:(9)$$A^{\pi }\left(s,a\right)=Q^{\pi }\left(s,a\right)-V^{\pi }\left(s\right)$$

Intuitively, subtracting value $V^{\pi }\left(s\right)$ (as opposed to an action-value $Q^{\pi }\left(s,a\right)$, shown in Equation (2), $V^{\pi }\left(s\right)$ refers to the state-value) as a baseline leads to smaller gradients which results in smaller updates, and this leads to stable learning. By combining Equations (7) and (9), we derive Equation (10), and by combining (8) and (9), we derive Equation (11):(10)$$\nabla_{\omega }L\left(\omega \right)=E\left[\nabla_{\omega }\log\pi_{\omega }\left(s,a\right)\;A^{\pi_{\omega }}\left(s,a\right)\right]A^{\pi }\left(s,a\right)=Q^{\pi }\left(s,a\right)-V^{\pi }\left(s\right)$$
(11)$$\omega_{j+1}=\omega_{j}+\alpha_{\pi }\;\nabla_{\omega }\log\pi_{\omega }\left(s,a\right)\;A^{\pi_{\omega }}\left(s,a\right)$$

In this work, we use A2C methods because of their advantages over policy-based and value-based methods.

## 4. Methodology

### 4.1. Agent Design

In this section, we discuss real-world constraints that need to be included in agent design, with background justification for each case. Additionally, we present our action, space, and reward definition.

**Constraints.** As discussed in the introduction, some real-world constraints need to be considered. Simulation is usually a ‘perfect world’ assumption where pedestrians are not part of the environment. If we do not maintain the following must-have constraints, the agent always tries to get a high reward; thus, it does not care about pedestrians or starved vehicles (i.e., vehicles that have been waiting for a long time). Note that even though these constraints conflict with maximizing the throughput, they need to be satisfied in order to make the model suitable for real-world applications.
In a real-world environment, the phase order in the traffic light is important for large intersections with pedestrian crossing sections. If the order of the traffic light is not preserved, pedestrians cannot cross the intersections safely. Additionally, some vehicles might end up waiting for a green light for a long time, which leads to *phase starvation*. Figure 2a shows an illustration of an intersection with four phases. When phase #2 is set (Figure 2b), pedestrians can cross the intersection through A–C and B–D directions; when phase #4 is set, they can cross through A–B and C–D.The minimum phase duration is necessary to enable safe pedestrian crossing. The minimum time required for safe crossing is determined by the width of the road in the intersection and the average walking speed of pedestrians. Generally, pedestrian walking speeds at the crosswalk in normal conditions range from 4.63 km/h to 5.37 km/h [56]. Thus, the minimum phase duration for roads, for example, with a width of 15 m and 20 m should be about 12 s and 15 s, respectively.Maximum phase duration is also necessary to prevent starvation of vehicles. Increasing the duration of the phase indefinitely, e.g., due to continuously incoming vehicles, may cause starvation in other lanes.

**State definition.** The state s is defined for each intersection i at time t as $s_{t}^{i}$. The state includes a current phase p, its duration τ, and the biased pressure (BP) of each phase. This means that our state definition does not depend on the number of traffic movements (e.g., 8, 12, 16), which gives flexibility on road network configuration and the advantage over existing MaxPressure methods, such as MPLight [25] and PressLight [39].

*MaxPressure*. The objective of the MaxPressure control policy is to maximize the throughput of the intersection by maintaining equal distribution in different directions, which is achieved by minimizing the pressure of the intersection. It is determined by the difference of queue length on incoming and outgoing lanes, as shown in Equation (12). However, MaxPressure control is not suitable for the cyclic control schemes. The problem with MaxPressure is that it sets the same pressure for different lanes if the difference between incoming and outgoing traffic is the same. Figure 3 shows an example scenario where the pressure values of phases #1 and #4 are both zero. Therefore, the MaxPressure-based agent treats them equally. However, phase #4 has more approaching vehicles. In cyclic control, any phase with the highest pressure cannot be selected unless it is the next phase. Therefore, lanes with more incoming vehicles (even though these lanes have the same pressure) require more time to free the intersection.
(12)$$P_{t}^{p_{k}}=\underset{\left(l,m\right)\in p_{k}}{\sum }\left[q_{t}^{l}-q_{t}^{m}\right]$$
where $P_{t}^{p_{k}}$ denotes the pressure of phase $p_{k}$ and $q_{t}^{l}$ and $q_{t}^{m}$ denote the queue length of incoming lanes l and outgoing lanes m associated with phase $p_{k}$, respectively.

*Biased Pressure (BP)*. To address the above-mentioned issue, we introduce a new version of MaxPressure control by putting some bias towards the number of approaching vehicles (or/and waiting vehicles) in the incoming lanes. To do this, we calculate the pressure of each phase and add the number of vehicles in the incoming lanes of this phase, as shown in Equation (13):(13)$$BP_{t}^{p_{k}}=\underset{l\in p_{k}}{\sum }N_{t}^{l}+\underset{\left(l,m\right)\in p_{k}}{\sum }\left[q_{t}^{l}-q_{t}^{m}\right]$$
where $N_{t}^{l}$ denotes the number of approaching vehicles in the incoming lanes l associated with phase $p_{k}$. Note that, unlike phase selection methods where the agent sets a phase with the highest pressure, the aim of our work is to determine the phase duration. Moreover, the limitation of cyclic control compared to non-cyclic control is that the phase order should be preserved, which means the agent cannot set the previous phase without completing the cycle. Therefore, adding the number of approaching vehicles, as shown in Equation (13), helps the agent select a more appropriate phase duration. Coming back to an example scenario shown in Figure 3, $BP_{p_{k}}$ values for phases #1 and #4 are no longer the same, which means an agent now has ‘more’ information about the traffic situation in the intersection to take a better action, e.g., the duration of phase #4 is set longer than phase #1. The complexity of BP is the same as MaxPressure but performs better for the cyclic control scheme, which will be justified in the experiments.

**Action definition.** As has already been discussed, we use cyclic signal control, in the same order shown in Figure 2b. Therefore, an agent only needs to select the phase duration τ for each phase p at time t as its action $a_{t}$ from the pre-defined action set A. To maintain a minimum phase duration constraint, we use an action set of {15, 20, 25, 30, 35, 40, 45, 50, 55, 60} for phases #2 and #4. This means that an agent selects the green light duration of mentioned phases from the given set. Since phase #1 and phase #3 do not involve any pedestrian crossing, we use an action set of {0, 5, 10, 15, 20, 25, 30, 35, 40, 45} for these phases. Note that the purpose of using separate action set for different phase groups is to maximize the throughput optimally (i.e., there is no need to wait for a minimum of 15 s if there is no traffic). The presence of {0} in the second set means the phase can be skipped if there is no traffic in the incoming lanes associated with the phase. Similarly, our method can be applied to non-cyclic phase control by introducing {0} in the action space for all phases. In that case, the agent sets the phase with the highest BP value. In addition, each green light of the corresponding phase is followed by 3 s yellow light. The specified phase durations and their range (e.g., {15, 20, 25, 30, 35, 40, 45, 50, 55, 60} and {0, 5, 10, 15, 20, 25, 30, 35, 40, 45}) in the action sets were determined through empirical methods and based on existing literature. Note that the duration step in the action set can be changed from 5 s to, for example, 3 s or 10 s, if necessary (additionally, action space can be continuous by setting the step to 1 s). For this work, 5 s was preferred because it is small enough to make a smoother transition and large enough to notice the action outcome so that the agent can be rewarded accordingly.

**Reward definition.** The reward is defined for the intersection, which means that an agent receives a reward based on the BP change on the whole intersection for taking an action. In this fashion, the agent is punished for the wrong selection of phase duration. The intuition behind the usage of intersection BP instead of phase BP is that every decision of an agent affects the distribution of incoming and outgoing lanes, which in turn, changes the distribution of vehicles in between several intersections. To increase the road network’s throughput, an agent needs to minimize the BP of the intersection. Therefore, the dynamics of intersections are more important than phase dynamics in a bigger picture. The reward $r_{i}\left(t\right)$ for the intersection i is calculated with Equation (14):(14)$$r_{t}^{i}=-BP_{t+∆t}^{i}=-\underset{p_{k}\in i}{\sum }BP_{t+∆t}^{p_{k}}$$
where $BP_{t+∆t}^{i}$ denotes the sum of BP of each phase associated with an intersection i at timestep t+∆t and ∆t denotes an interaction period between the agent and environment.

### 4.2. Framework Design and Training

**Framework design.** The overall architecture of our proposed deep RL model is shown in Figure 4. From the environment, each agent i observes state $s_{t}$ of its own intersection at time t and depending on the current phase p and traffic state $s_{t}$, each agent takes an action $a_{t}$, which results in state change $s_{t+1}$, and receives reward $r_{t}$ from the environment. After timestep u, the network parameters are updated.

The structure of the deep neural network (DNN) used in this paper is also shown in Figure 4. We use long short-term memory (LSTM) after two fully connected layers. The purpose of using LSTM in the network is to help the model memorize the short history (also called temporal context) of state representation. LSTM has been used widely in recent RL applications because it has the ability to keep hidden states [34,57]. The LSTM layer is followed by the output layer, which consists of two parts, critic and actor.

Update functions for critic and actor are shown in Equations (15) and (16), respectively:(15)$$\theta_{j}^{i}=\alpha \;{𝛻}_{\theta }\left(\frac{1}{2B}\underset{t\in B}{\sum }{\left(Q_{\theta }^{i}\left(s_{t},a_{t}\right)-V_{\theta }^{i}\left(s_{t}\right)\right)}^{2}\right)$$
(16)$$\omega_{j}^{i}=\alpha_{\pi }\;{𝛻}_{\omega }\left(\frac{1}{B}\underset{t\in B}{\sum }\log\pi_{\omega }\left(s,a\right)\;A_{t}^{\pi_{\omega },i}\left(s_{t},a_{t}\right)\right)$$
where B is a minibatch and contains experience trajectory $<s_{t},\;a_{t},\;s_{t+1},\;r_{t}>$, and discounted reward function $Q_{\theta }^{i}\left(s_{t},a_{t}\right)$ is derived from Equation (2), in which $r_{t}^{i}$ is derived from Equation (14).

**Training**. We train our model from scratch using a trial-and-error mechanism. To increase the efficiency, the model (a single agent) is first trained on a single intersection for 30 episodes, with each episode lasting for 30 min. Then, this agent is duplicated for all intersections of the given road network (multi-agent) and trained for another 40 episodes. Note that such training approach preserves computational resources because in the early stages, an agent only needs to learn the correlation between traffic volume and its movement with the corresponding appropriate actions. Therefore, training a single agent first and then sharing its gained ‘common knowledge’ with other agents is favorable.

**Network hyperparameters.** Hyperparameters of DNN and the training process are finetuned using the grid search. Fully connected layers of the DNN, shown in Figure 4, have 6 and 64 nodes, respectively, and an LSTM layer has 32 nodes. A softmax layer with 6 nodes is used for the actor and a linear layer is used for the critic. The parameters θ and ω of the critic and actor are then trained using Equations (15) and (16) with their corresponding learning rates $\alpha =10^{-3}$ and $\alpha_{\pi }=10^{-4}$. Minibatch size B = 64 is used to store the experience trajectories. The discount factor γ is set to 0.99.

## 5. Experimental Environment

Our experimental evaluation has two major objectives. The first objective is to justify the superiority of the proposed BP method. We demonstrate it through ablation studies using BP and the existing pressure methods, such as MaxPressure [32] and PressLight [39]. The second objective is to show the effectiveness of the proposed model. We conduct comprehensive experiments on both synthetic and real-world datasets and road network structures to compare our method with the existing cyclic control methods. We implemented our method using the Keras framework with a Tensorflow backend. Experiments are conducted with the NVIDIA Titan Xp graphics processing unit. The experiments are conducted on CityFlow (https://cityflow-project.github.io/, accessed on 2 February 2022), an open-source large-scale traffic control simulator that can support different road network designs and traffic flow [58]. Derived synthetic and real-world datasets are designed to fit the simulation. CityFlow includes API functions to derive the number of vehicles in the incoming and outgoing lanes. For real-world situations, this information can be obtained by smart sensors or cameras.

### 5.1. Metrics for Performance Evaluation

To evaluate and compare the performance of different methods, we use two types of evaluation metrics:**Travel time**. The average travel time of vehicles is measured in seconds. This metric is widely used by research works in the transportation field and traffic light control. It is calculated by dividing the travel time of all vehicles by the number of vehicles in the last two episodes. Shorter travel time is better in comparison. However, this metric alone cannot determine which traffic plan is better. For example, if the road is full due to a bad signal plan, then incoming cars cannot join the road network. Therefore, we also use throughput as a metric.**Throughput**. We denote the closed road network in the simulation as “city”. Then, throughput is measured by the number of vehicles that have left the city in the last two episodes of testing. For example, as shown in Figure 5, four vehicles have left the city between time step t and t+1. We sum the number of leaving vehicles during the whole two-episode duration to calculate throughput. Higher throughput is better in comparison.

### 5.2. Datasets

We use synthetic and real-world datasets to evaluate the performance of our work and related methods. Both synthetic and real-world datasets are adjusted to fit the simulation settings.

**Synthetic data.** Four configurations are used for a synthetic dataset, as shown in Table 1. To test the flexibility of the deep RL model, both normal and rush-hour traffic (denoted as light and heavy traffic, respectively) flows are used.

**Real-world data.** For the real-world scenarios, we use New York and Seoul transport data on vehicle trip records and their corresponding road networks. Figure 6 shows road networks for New York and Seoul. New York vehicle trip record data are taken from open-source trip data (https://www1.nyc.gov/site/tlc/about/tlc-trip-record-data.page, accessed on 23 November 2021). To use these data in the CityFlow simulator, we map each vehicle’s origin and destination from the geo-locations of the data. In the same fashion, we create a dataset from Seoul transport data (https://kosis.kr/eng/statisticsList/statisticsListIndex.do, accessed on 23 November 2021). To complete the dataset, we use a combination of four statistical record databases: Vehicle Kilometer Statistics, Traffic Volume Data, Road Statistics, and National Traffic Survey. The dataset information is roughly summarized in Table 2.

Although the number of intersections varies significantly between two road networks (New York’s 90 against Seoul’s 20), the areas of selected regions are almost the same (~1.8 km^2^).

### 5.3. Compared Methods

To evaluate the performance of our model, we use the following related methods. In this work, we use related cyclic control methods that use phase duration selection (not phase selection) as their action definition since phase order is important in most real-world intersections, as explained in Section 4.1. We also demonstrate the performance difference between cyclic and non-cyclic designs.

4.Fixed-time signal plan with GreenWave effect (FT). This is the most commonly used technique in real-world traffic light control systems. The cycle length, offset, and phase duration of each intersection is pre-determined based on historical data of the intersection. In this paper, we use 30 s and 40 s FT approaches, where FT 30 s and FT 40 s mean that the duration of green light in a phase is 30 s and 40 s, respectively.5.MaxPressure [32]. This method uses queue length to represent the state and greedily selects the phase which minimizes the pressure of the intersection. By doing so, it aims to maximize the throughput of the intersection, and ultimately, the throughput of the whole network. MaxPressure method is selected for comparison because it is widely used on the recent state-of-the-art methods, such as MPLight [25] and PressLight [39].6.BackPressure [35]. This method is an adaptation of MaxPressure for the cyclic scheme. In the beginning, cycle duration is fixed, and the duration of each phase is determined proportionally, depending on the pressure of each phase in the intersection.7.3DQN [21]. This method is the combination of DQN, Double DQN, and Dueling DQN. State representation is based on the image of vehicle positions in the incoming lanes. The authors divided the whole intersection into small grids and used a matrix to represent the position of each vehicle. The action is whether to increase or decrease the phase duration by 5 s. This is one of the state-of-the-art methods that adjusts the duration of each phase depending on the traffic, and this method beats previous works on DQN methods, such as [19] in terms of travel time.

## 6. Numerical Results

### 6.1. Ablation Study

In this section, we present ablation studies on the proposed Biased Pressure (BP) to justify the action and state definitions of our RL agent. In the first experiment, we compare the performance of the BP coordination method with MaxPressure. Additionally, we present the results of Fixed-time 30 s signal plan as a reference point. Since the purpose of this experiment is to show the effectiveness of the BP method, we use the same model with the same network structure for both methods. The experiment procedure is repeated using both $RN_{3\times 3}$ and $RN_{4\times 8}$ road networks with four configurations of traffic volume from Table 1. Methods are compared in terms of the average travel time of vehicles within the network during the first 100 min of simulation. Results are shown in Figure 7.

In each configuration, our method outperforms the existing MaxPressure method. For the $RN_{3\times 3}$ road network, the proposed BP method achieves, on average, a 10–15 s improvement in travel time. The performance gap is especially seen with the $RN_{4\times 8}$ network, Figure 7b,d, where BP achieves about 60–70 s shorter travel time for each vehicle, compared to MaxPressure. The reason why the BP method works better than the conventional MaxPressure method is that our model has more information about the traffic situation and, thus, selects a more appropriate phase duration, e.g., a longer phase if there are more incoming cars and a shorter phase if the opposite, given that the pressure of the lanes is the same. It is important to note that agents can take actions (i.e., phase duration selection for the next phase) within 3–5 s in our local machine, which facilitate real-time signal optimization.

In the second experiment, we compare the performance of cyclic and non-cyclic BP methods. Cycle-based approaches keep the order of phase and are more suitable for city-level traffic signal control. However, non-cyclic approaches can select any phase with the highest BP value, and therefore, they can quickly react to the traffic flow, which helps to maximize the throughput. Table 3 shows a comparison between cyclic and non-cyclic BP methods. Due to flexibility, non-cyclic BP outperforms cyclic BP in all configurations of synthetic data. The performance gap is larger for Config 3 and 4 in terms of travel time and throughput, which means non-cyclic BP excels in heavy traffic. This ablation study is deliberately conducted to validate that our proposed BP method can be applied in both cyclic and non-cyclic phase control schemes. Moreover, the comparison between our BP method and PressLight method is demonstrated for non-cyclic control. For Config 1 and 2, PressLight outperforms the BP method. Moreover, for Config 3 and 4, which represent a heavy traffic flow, our method achieves better results.

However, as explained in previous sections, the motivation of this research is to propose an efficient model for real-world situations, where, unlike simulations, intersections also include pedestrian traffic and require a cyclic phase. Therefore, the remaining experiments include only cyclic control schemes.

### 6.2. Performance Comparison

**Synthetic data.** In the first experiment, we compare the performance of the related methods on a synthetic dataset. The average travel time and throughput of each method are measured for all configurations to test the flexibility of methods with different traffic volumes. Results are collected from the last hour of simulation. Table 4 shows the performance comparison of different methods on the $RN_{3\times 3}$ road network. Our method outperforms all related methods with a notable margin. Travel time is calculated for each vehicle; therefore, the numerical difference is not significant. The greatest marginal difference between our method and the second-best method (of the corresponding Config) is 19.8 s and it is seen in Config 2, and the smallest marginal difference is achieved in Config 4 with 8.3 s travel time improvement. When it comes to throughput, our method allows 120 more vehicles to pass in each hour on average with light traffic (Config 1 and 2) and 334 more vehicles with heavy traffic (Config 3 and 4). This means that 8016 more vehicles can pass the 3 × 3 road network on a daily basis with our traffic signal control model, which is significant.

We repeat the same experiment on the $RN_{4\times 8}$ road network. This network is larger than $RN_{3\times 3}$ and has more intersections. Moreover, the number of SN and WE directions is different which creates different traffic fluctuations and imbalance (due to SN and WE asymmetry) in the network. Results of compared methods are shown in Table 5. Our method outperforms the related methods in all traffic scenarios. In terms of travel time, our method achieves an average of 47.7 s improvement in all configurations. In Config 1, for example, vehicles complete their trip 61.4 s faster than the second-best method, which is BackPressure. The performance gap shown in Table 5 is significantly higher than in Table 4, meaning that our method dominates over related works more and more, as the road network size gets bigger and bigger. A similar trend is seen in terms of throughput. In Config 3, 478 more vehicles can pass the network each hour with our method, making it an average of 11,472 more vehicles daily.

According to the overall numerical results, FT 40 s and FT 30 s achieve the worst performance in comparison because they use fixed duration for all phases regardless of the traffic situation. MaxPressure, BackPressure, and 3DQN methods perform relatively equally, outperforming each other in the respective configurations. However, MaxPressure shows more robust performance in all traffic scenarios. The limitation of BackPressure is that the method uses fixed cycle length, which is less flexible to changing traffic, whereas the limitation of 3DQN is that it changes the traffic signal by 5 s, which results in a slower reaction to dramatic changes in the traffic.

**Real-world data.** In this part of the experiment, we compare the performance of each method using real-world datasets and their corresponding road networks. Table 6 shows numerical results for New York and Seoul datasets. Although the number of intersections is greater in the New York network, Seoul has a heavier traffic volume, as discussed in Section 5.2. Our method shows superior performance in both datasets, with the least travel time and the highest throughput. In the New York dataset, 3DQN achieves the second-best result with a travel time of 189.30 s, compared to our method’s 152.90 s. In the Seoul dataset, our method achieves 123.48 s travel time, whereas the second-best result is obtained by MaxPressure, with 131.29 s. The throughput difference between our method and the second-best method (in the corresponding dataset) for New York and Seoul datasets is 257 and 244, respectively. From the overall trend, we can see that our method performs better as the number of intersections increases. This means that our method shows more scalability potential, owing to simple and well-designed state, reward, and action definitions.

## 7. Conclusions

In this paper, we propose a multi-agent RL-based traffic signal control model. We introduce a new BP method, which considers not only the phase pressure but also the number of incoming cars to design our state representation and reward function. Additionally, we consider real-world constraints and define our action space to facilitate cyclic phase control with minimum and maximum allowed duration for each phase. Setting different action spaces for different phases proves to maximize the throughput while enabling safe pedestrian crossing. We validate the superiority of the proposed BP over the existing pressure methods through an ablation study and experiments. Experimental results on synthetic and real-world datasets show that our method outperforms the related cycle-based methods with a significant margin, achieving the shortest travel time of vehicles and the highest overall throughput. Robust performances in different configurations of traffic volume and various road network structures justify that our model can scale up and learn favorable policy to adjust appropriate phase duration for traffic lights.

Even though our method produced promising results, there is a need for future improvements in intelligent traffic signal control systems. Results should be derived using real traffic lights instead of simulated traffic lights to evaluate the authenticity of RL-based methods, since real-world traffic situations include a dynamic sequence of traffic movements. Impatient drivers and pedestrians, drivers’ mood, their perceptions, and other human factors as well as the involvement of long vehicles, such as trucks and busses, play a great role in decision making, and ultimately, affect traffic movement and dynamics. Moreover, computational complexity and feasibility of decision making at the real-world intersections should be investigated, since obtaining data via smart sensors or cameras costs more than via numerical data used in simulations. Therefore, more real-world constraints should be studied, and the reality gap between the real world and simulated environment should be examined thoroughly.

## Figures and Tables

**Figure 1 sensors-22-02818-f001:**
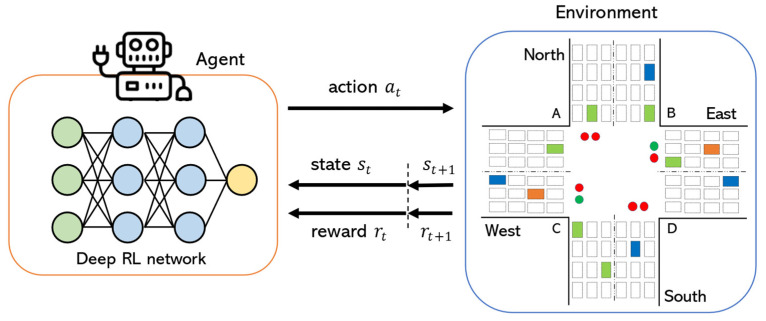
Deep reinforcement learning model for traffic light control.

**Figure 2 sensors-22-02818-f002:**
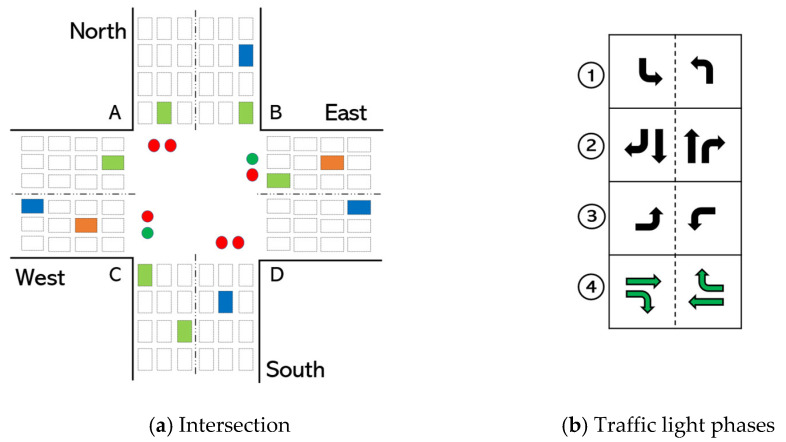
The illustration of (**a**) a road intersection that has four directions and twelve traffic movements, and (**b**) four traffic signal phases. Here, phase #4 is set for an intersection. The phase order is set cyclic as #1-2-3-4-1-2, etc.

**Figure 3 sensors-22-02818-f003:**
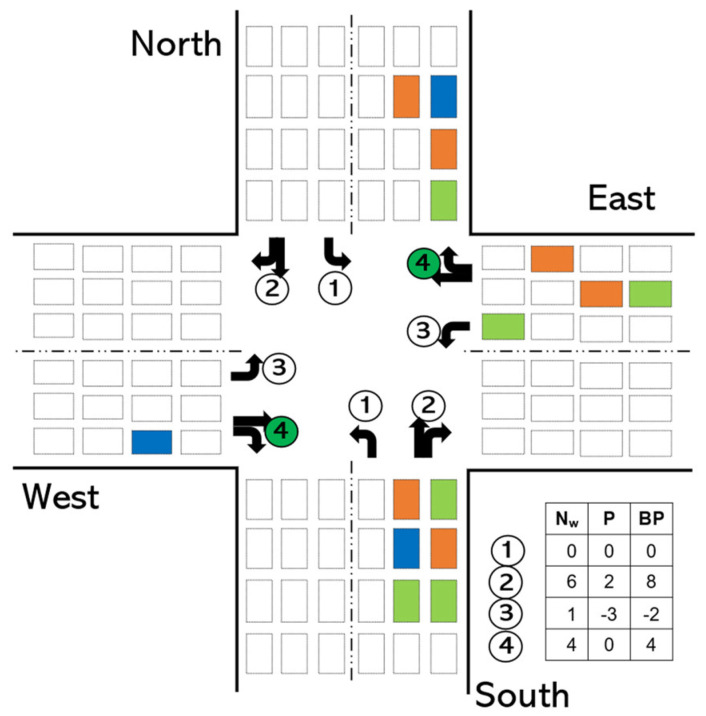
Example scenario to illustrate the difference of conventional pressure control and the proposed BP control. Phase #4 is set for the intersection. The table on the bottom-right corner of the figure shows the sum of approaching vehicles in the incoming lanes (denoted as N_w_), pressure value (denoted as P), and BP value (denoted as BP) for all four phases.

**Figure 4 sensors-22-02818-f004:**
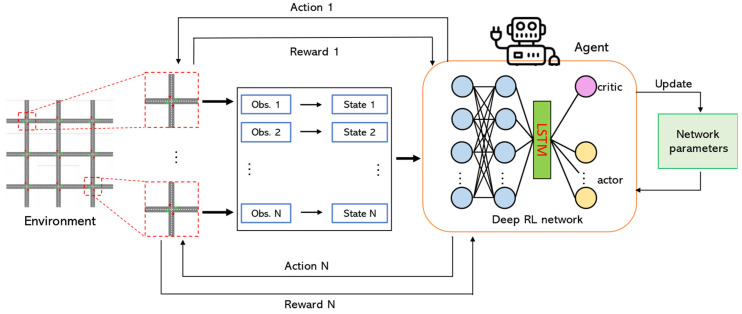
The overall framework of the proposed method. Each deep RL agent controls its own intersection in the environment.

**Figure 5 sensors-22-02818-f005:**
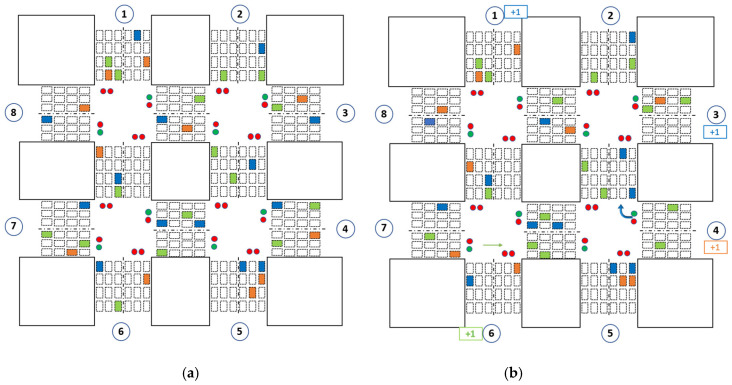
The traffic state of the road network, also called a city, (**a**) Traffic network state at time t and (**b**) Traffic network state at time t+1. As seen, four vehicles have left the city at the destinations labeled as 1, 3, 4, and 6. Thus, throughput is four between t and t+1.

**Figure 6 sensors-22-02818-f006:**
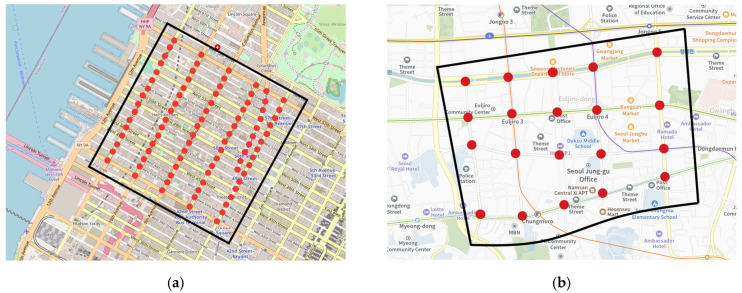
Road network for real-world data: (**a**) Part of Hell’s Kitchen, Manhattan, New York, USA and (**b**) Central area of Jung-gu, Seoul, South Korea. For both road networks, the coverage of the monitored area is marked with black lines, and red points represent corresponding intersections in the area.

**Figure 7 sensors-22-02818-f007:**
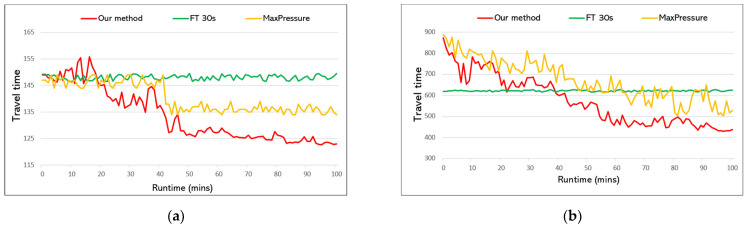
Performance comparison between proposed Biased Pressure (denoted as ‘Our method’), MaxPressure, and Fixed-time 30 s (‘FT 30 s’) using $RN_{3\times 3}$ and $RN_{4\times 8}$ road networks with four configurations of traffic volume. Our method shows better performance in each configuration. (**a**) $RN_{3\times 3}$ with Config 1. (**b**) $RN_{4\times 8}$ with Config 3. (**c**) $RN_{3\times 3}$ with Config 4. (**d**) $RN_{4\times 8}$ with Config 2.

**Table 1 sensors-22-02818-t001:** Various configurations of synthetic dataset.

Config	Directions	Arrival Rate (vehicles/min)	Traffic Volume
1	All	8	Light(normal traffic)
2	NS & SN	6
EW & WE	10
3	All	15	Heavy (rush hours)
4	NS & SN	12
EW & WE	18

**Table 2 sensors-22-02818-t002:** Summary of two real-world datasets.

Dataset	Arrival Rate (Vehicles/min)				Number of Intersections
	Mean	Std	Max	Min	
New York	46.12	3.42	53	40	90
Seoul	82.90	6.34	98	71	20

**Table 3 sensors-22-02818-t003:** Performance comparison of cyclic and non-cyclic phase control on the $RN_{3\times 3}$ network within one hour of simulation. Best results are shown in bold.

Type	Travel Time				Throughput			
	Config 1	Config 2	Config 3	Config 4	Config 1	Config 2	Config 3	Config 4
Cyclic BP	112.69	123.10	182.47	179.93	4617	4508	9292	9356
Non-cyclic BP	107.14	119.76	**163.91**	**158.71**	4739	4632	**9415**	**9498**
PressLight	**106.21**	**118.80**	168.66	163.13	**4751**	**4645**	9384	9412

**Table 4 sensors-22-02818-t004:** Performance comparison of different methods on the $RN_{3\times 3}$ network.

Methods	Travel time				Throughput			
	Config 1	Config 2	Config 3	Config 4	Config 1	Config 2	Config 3	Config 4
FT 30 s	147.79	153.78	195.34	199.10	4298	4271	8680	8805
FT 40 s	155.23	159.41	200.74	201.94	4127	4236	8548	8690
MaxPressure	131.58	142.93	194.59	188.21	4490	4395	8971	9010
BackPressure	143.73	146.90	193.64	190.37	4285	4229	8730	8867
3DQN	141.47	150.28	191.16	192.80	4331	4290	8863	8733
Our method	112.69	123.10	182.47	179.93	4617	4508	9292	9356

**Table 5 sensors-22-02818-t005:** Performance comparison of different methods on the $RN_{4\times 8}$ network.

Methods	Travel time				Throughput			
	Config 1	Config 2	Config 3	Config 4	Config 1	Config 2	Config 3	Config 4
FT 30 s	417.23	398.89	660.81	611.63	7163	7249	9570	10179
FT 40 s	430.28	414.81	612.61	601.27	7097	7118	9732	10319
MaxPressure	313.94	284.54	474.83	480.84	8219	8361	10113	11070
BackPressure	325.92	280.17	501.80	481.04	7592	8314	9920	11021
3DQN	328.87	283.35	484.93	491.90	7931	8210	10187	10896
Our method	252.58	246.74	436.92	422.56	8412	8552	10665	11427

**Table 6 sensors-22-02818-t006:** Performance comparison of different methods on real-world data.

Methods	Travel Time		Throughput	
	New York	Seoul	New York	Seoul
FT 30 s	196.25	162.03	2119	4248
FT 40 s	210.61	174.71	2087	4190
MaxPressure	195.42	131.29	2345	4455
BackPressure	202.78	132.85	2281	4492
3DQN	189.30	137.68	2316	4331
Our method	152.90	123.48	2602	4736

## Data Availability

The private data presented in this study are available on request from the first author. The publicly available datasets can be found at https://www1.nyc.gov (accessed on 23 November 2021) and https://kosis.kr (accessed on 23 November 2021).

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
