# Peer review of "Biased Pressure: Cyclic Reinforcement Learning Model for Intelligent Traffic Signal Control"

_sensors, 2022, doi:10.3390/s22072818_

Round 1
Reviewer 1 Report
The review ot deep learning model can be shoretened as it is nto the centre of the research.
How You see your model working with intersections that have other configurations/models (e.g. static light cycle). Will it fit?
Did You measure teh throughtput of the pedestrians and their waiting time?
Author Response
We look forward to your next revision. Thank you for your time. Please refer to the attached revision file.

Reviewer 2 Report
The manuscript is well-organized and written. The methodology based on reinforced learning (RL) is sound for traffic light control. It is somewhat novel to employ BP for improving the performance in comparison with some other works use pressure methods. The experimental tests are sound, and they are interesting to readers. Accordingly, the merits of the manuscript warrants "Accept as it is".
Author Response
Thank you for your positive review. We appreciate your time.

Reviewer 3 Report
This paper considers multi-intersection traffic light control problem. The paper propose a multi-agent RL-based traffic signal control model and uses a Biased Pressure (BP) method. The paper use two types of evaluation metrics, travel time and throughput, to evaluate and compare the performance with different methods. Experimental results show that this method has optimization effect. Overall it is a good study and does propose a solution to multi-intersection traffic light control. However, I still have some concerns regarding the formulation and simulation experiment.
1 From Equation (12) and formula (13), we know the Biased Pressure control policy is an improvement of the Max Pressure control policy. The Max Pressure control policy is a decentralized adaptive traffic signal control algorithm, which has been proved to be scalable and stable, but this method requires real-time information of queue length and turning rate, the improved model may also need the information of the queue length and turning rate in real-time. However, the determination method of queue length and turn rate is not introduced in detail in this paper. Please supplement.
2 Queue length is rarely discussed in this paper. When the queue length is uneven at different intersections, how will the optimization effect of the proposed model.
we calculate the pressure of each phase and add the number of vehicles in the incoming lanes of this phase
3 Why you add the number of vehicles in the incoming lanes of this phase to MaxPressure in Equation (12)? Is it reasonable? What is the rationale behind the addition?
- As we all know, in real-time optimization, the calculation efficiency of the model is a very important problem. The article does not explain the time spent in optimizing the signal timing each time. Please supplement.
Author Response

(The authors gave the same response as above.)

Reviewer 4 Report
• The authors propose a model of multi-intersection traffic light control based on reinforcement learning with a simple yet effective combination of state, reward, and action definitions in this article. The proposed model makes use of a novel pressure method called Biased Pressure (BP) and distributes the developments to increase scalability. In general, this paper is well-written, and the subject is intriguing. Here are my comments about this work.
• The notion of incorporating real-world constraints into the DRL-based framework is an excellent one. It is suggested that the authors clarify whether there are any implicit or explicit assumptions or constraints when adding such constraints. In other words, it would be beneficial if the authors could emphasize the scenarios or use cases in which such constraints are applicable to this model.
• As the DRL-based model is quite distinct from conventional traffic light control models used in transportation engineering. It would be interesting to see if the DRL results are interpretable. It would be beneficial if the authors could evaluate the obtained offsets, green waves, and traffic light coordination across the network.
Author Response
We look forward to your positive response. Thank you for your time. Please refer to the attached revision file.

Reviewer 5 Report
The authors propose a methodology for designing intelligent traffic lights.
Although the topic is of great interest, the paper has several flaws that affect the paper's goodness. Indeed, the authors provide a very superficial literature review neglecting numerous contributions and analysing very historical ones such as SCOOT (1982), SCATS (1992), Miller (1963) and Maxband (1981).
Obviously, in the last 20-30 years something has been done. For example, there were proposed some design models indicated “in the literature” as asymmetric traffic assignment models where in the first decade of the 2000s the researchers have developed models since the variation in the timing of a traffic light produces a variation in the travel times of the users and therefore in the distribution of flows on the networks. But since the design of a traffic light requires the knowledge of the flows it is necessary to consider jointly the flows that influence the timing and the timing that influences the flows in an asymmetrical way (i.e., the Jacobian of the cost functions is not symmetrical).
Obviously, there are dozens of contributions (currently become fundamental in the literature) that the authors have completely neglected: for example the use of Artificial Neural Networks, Nature Inspired Solution Algorithms (Ant Colony Optimization approaches, Honey Bee approaches, etc.) or simply the Exact Methods category.
Neglecting the fundamental literature of the past 20-30 years has produced:
- The adoption of completely false sentences even in the abstract (see, for instance, the sentence “existing methods [… omissis …] neglect the real-world constraints such as cyclic phase order and minimum/maximum duration for each traffic phase”).
- The development of an application in which the proposed methodology is compared in the case of 2 networks (New York and Seoul) with other methods and surprisingly the authors’ proposal always shows better results (dominant methodology compared to the others).
Among other things, I would have expected a comparison in terms of performance (algorithm complexity), calculation times or reduction of congestion compared to methods currently adopted in the design and planning of real traffic lights.
For all the above-mentioned reasons, I consider the contribution not eligible for publication.
Minor observations
Please verify the citation order since in the text there is no progressiveness in the cited contribution.
Author Response

(The authors gave the same response as above.)

Round 2
Reviewer 5 Report
In the previous review round, I had expressed my perplexities with respect to a methodology that proposed a state of the art of the 80s and 90s, completely neglecting the research that has been carried out in the last 20-30 years in the field of simulation and design/planning of traffic lights devices.
The completely superficial analysis of the literature had implied the presence of completely false statements in several places in the paper.
Among other things, the application showed the methodology proposed by the authors as the one with dominant improvements compared to all the other methodologies proposed in the literature (but which literature ??).
Despite my observations and my perplexities in considering the paper admissible for publication, the authors in the new version drawn up in a couple of days (my revision was March 1 while the new paper is March 5) consists in having added 3 references (the purpose of a state of the art is to analyse the contents of the literature and not simply to cite the contribution) and to have modified/added 4 sentences (page 2, page 11, page 15 and page 17).
For all of the above, I consider the contribution not admissible for publication.
Author Response
Please refer to the attached file. We look forward to a positive review.
